# Setting the top 10 priorities for obesity and weight-related research (POWeR): a stakeholder priority setting process

Ailsa R Butler [ID],[1] Nerys M Astbury [ID],[1] Lucy Goddard,[1] Anisa Hajizadeh,[1] Philippa Seeber,[2] Bruce Crawley,[2] Paul Aveyard [ID],[1] Susan A Jebb[1]

[1]Nuffield Department of Primary Care Health Sciences, University of Oxford, Oxford, UK
[2]Patient and Public contributor, N/A, UK

**Correspondence to**
Nerys M Astbury;
nerys.astbury@phc.ox.ac.uk

## ABSTRACT

**Objectives** To identify and prioritise the most impactful, unanswered questions for obesity and weight-related research.

**Design** Prioritisation exercise of research questions using online surveys and an independently facilitated workshop.

**Setting** Online/virtual.

**Participants** We involved members of the public including people living with obesity, researchers, healthcare professionals and policy-makers in all stages of this study.

**Primary outcome measures** Top 10 research questions to be prioritised in future obesity and weight-related research.

**Results** Survey 1 produced 941 questions, from 278 respondents. Of these, 49 questions held satisfactory evidence in the scientific literature and 149 were out of scope. The remaining 743 questions were, where necessary, amalgamated and rephrased, into a list of 149 unique and unanswered questions. In the second survey, 405 respondents ranked the questions in order of importance. During the workshop, a subset of 38 survey respondents and stakeholders, agreed a final list of 10 priority research questions through small and large group consultation and consensus. The top 10 priority research questions covered: the role of the obesogenic environment; effective weight loss and maintenance strategies; prevention in children; effective prevention and treatment policies; the role of the food industry; access to and affordability of a healthy diet; sociocultural factors associated with weight; the biology of appetite and food intake; and long-term health modelling for obesity.

**Conclusions** This systematic and transparent process identified 149 unique and unanswered questions in the field of obesity and weight-related research culminating in a consensus among relevant stakeholders on 10 research priorities. Targeted research funding in these areas of top priority would lead to needed and impactful knowledge generation for the field of obesity and weight regulation and thereby improve population health.

## BACKGROUND

Obesity is a major preventable cause of ill health and is affecting an increasing number of children and adults globally.[1][2] Obesity is defined as a body mass index (BMI) of ≥27.5 kg/m² (or ≥30 kg/m² if of white ethnic groups). No country has managed to

## STRENGTHS AND LIMITATIONS OF THIS STUDY

⇒ This is the first research priority setting specific to the field of obesity and weight-related research.
⇒ This exercise involved input from a large number of participants from a broad range of relevant stakeholder groups including patients, members of the pubic, researchers, policy-makers and charities.
⇒ The final list of priorities was reached through consultation and consensus in a workshop guided by independent facilitators to minimise bias towards certain questions or areas.

achieve a sustained decrease in the prevalence of obesity, despite evidence-based clinical and public health guidelines and polices aimed at tackling obesity.[3][4] Obesity increases the risk of developing several conditions including type 2 diabetes, cardiovascular disease, osteoarthritis and some cancers.[4] The cost attributable to overweight and obesity are substantial. For example, in the UK's National Health Service the cost is projected to reach £9.7 billion per annum, with wider costs to society projected to reach £49.9 billion by 2050 per year.[5][6] The detrimental effects of excess weight are not restricted to those who meet the BMI threshold of obesity as the increased morbidity is seen in people with any degree of excess adiposity.[7] Accordingly, strategies to prevent obesity or excess weight or adiposity are needed, defined here as obesity and weight-related research.

Presently, the research agenda is mainly driven by the interests and concerns of researchers, or research commissioners. A more transparent, systematic and collaborative approach involving multiple stakeholders to identify research priorities could accelerate progress. The James Lind Alliance (JLA) priority setting process brings patients, carers and clinicians together on an equal basis to define uncertainties, consider their importance and thereby set research priorities.[7][8] The output should, and typically has,

**BMJ**

informed researchers and research funders about the key questions to address in research because it is based on what matters most to people with lived experience of having a condition and those treating it.[9] Policies to prevent obesity typically affect the whole of society, for example fiscal policies or policies restricting the promotion or selling of some goods. Likewise, providing treatment for obesity as part of publicly funded healthcare is contested, and thus questions about research in this area seem to call for a much wider group of stakeholders than patients, carers, and clinicians. As in a previous tobacco control priority setting partnership (PSP),[10] we adapted the JLA approach to incorporate the perspectives of this wider range of stakeholders including people without experience of obesity, policy-makers, charities and, as for JLA, patients and members of the public with a lived experience of obesity (or related disease) and clinicians.[10] The objective of this work, as the first prioritisation project in obesity and weight-related research, was to identify unanswered questions across the whole of the field, from basic science through to health policy.

## PATIENT AND PUBLIC INVOLVEMENT

We involved two members of the public (BC and PS) with lived experience of overweight in all stages of the project, from conception and design of the study, to its conduct, data collection and analysis. Our wider public involvement (surveys and workshop) incorporated members of the public with and without lived experience of being overweight and patients, defined as people with lived experience of being overweight and experience of receiving clinical treatment for overweight, obesity or an associated condition. Members of the public were involved in all stages of the work alongside and as equal partners with other stakeholders.

## SUBJECTS AND METHODS

The priorities for obesity and weight-related research (POWeR) project took place between December 2019 and December 2020. The process was guided by Viergever *et al*,[11] which outlines principles of priority setting in health research and by the priority setting process carried out by Lindson *et al*.[11] We were guided by the general principles of the JLA PSP, however, we involved a wider range of stakeholders. We engaged a diverse and representative group of stakeholders comprising members of the public, people with lived experience of overweight and/or obesity, and people who work for organisations and charities, funders, policy-makers, clinicians and academic researchers all involved in the field of overweight and obesity. The prioritisation process had three stages: first an online survey to collect research questions stakeholders deemed to be priorities; a second online survey to rank the priority questions amalgamated from survey 1; and finally, an online workshop to reduce the ranked questions from survey 2 to produce a final list of top 10

priorities. The scope was limited to research questions on the aetiology, consequences, prevention or treatment of overweight and obesity in both adults and children, and did not include questions about whether currently evidenced interventions or polices should be implemented. For example, research questions relating to the prevention or treatment of eating disorders were not within the scope of this prioritisation project, however, eating disorders and related psychological adverse events related to weight management programmes were within scope. There was a study management group of investigators and patient and public involvement representatives that met regularly. Survey respondents provided consent to survey 1 and survey 2. Participants in the online workshop gave explicit consent prior to participation.

### Survey 1: gathering questions and identifying those unanswered

Gathering questions:

The first survey in the process asked respondents to submit up to four questions that they felt should be prioritised in the field of obesity and weight-related research. The survey was administered online using JISC Online Survey and was piloted with our public coauthors and colleagues in the research team, prior to being launched. The survey asked respondents to identify research questions that they felt were the most important unanswered questions on the topic and to say why they felt each question was important. The latter information was used by the team to interpret, contextualise, group and sort questions.

We publicised the survey passively via a web link on our POWeR project website (https://www.phc.ox.ac.uk/research/participate/power), and actively via email to relevant stakeholders and Facebook adverts targeted to men. We invited our stakeholders to circulate the link, resulting in the distribution of the survey by more than 40 organisations to their members, visitors to their webpages and readers of their newsletters (online supplemental table S1). We targeted organisations relevant to the field which included but was not limited to, obesity charities, community groups, funding bodies, hospital trusts, general practices and city councils. Participation in survey 1 was incentivised through a prize draw. We made physical copies of the survey, and versions with a large font size readily available on request. The survey was only available in English, and open for responses for 37 days between 15 January 2020 and 21 February 2020.

Identifying unanswered questions:

Survey 1 questions were grouped by topic area and rephrased to form answerable research questions (online supplemental tables S2 and S3). We used a multilevel coding system to categorise questions into overarching categories that were iteratively deduced throughout the grouping. For example, the submitted question 'which diets work' fell into a macro category, 'treatment' and was then further filtered into the sub-category 'behavioural' over 'pharmaceutical'. Questions organised into groups

were then rephrased as research questions in collaboration with our public coauthors who ensured that the groupings and rephrasing retained the intent of the original questions, and that they were understandable to a lay audience while making them tractable to empirical research. For example, a question such as 'Are there medications to treat obesity?' would have been combined with others to become a tractable research question such as 'What is the effectiveness, safety, tolerability and cost-effectiveness of medications to treat obesity?'

We then searched the literature using keywords and MeSH terms informed by the questions, to determine if these were areas that were already adequately addressed in the scientific literature. Questions were deemed 'answered' if there was satisfactory evidence. We accepted satisfactory evidence primarily in the form of preprocessed literature in: (1) systematic reviews published within the last 10 years, with little to no uncertainty; (2) proof of evidence in national clinical guidelines (eg, National Institute for Health and Care Excellence, and Scottish Intercollegiate Guidelines Network). We also accepted primary literature by way of high certainty if there was evidence in randomised controlled trials (RCTs). Such an approach would indicate that the question on currently available pharmacotherapy for obesity, for example, was at least partially answered by current reviews and trials.

We noted how many questions fed into each research question.

## Survey 2: prioritising unanswered research questions

The second online survey was piloted with members of the public and colleagues in the research team. The survey was administered via REDCap, and sent to the 256 survey 1 respondents who had provided us with their email addresses, as well as to the organisations approached to share survey 1 (see online supplemental table S1).

The second survey remained open for 30 days between 6 August 2020 and 14 September 2020. Survey 2 asked respondents to prioritise the unanswered questions gleaned from survey 1, which were sent in batches of about 50 questions to lower the response burden. The questions in each batch covered the whole range of submitted research questions. Respondents were asked to rate each question on a scale of 1–10 with 10 representing 'very important' and 1 representing 'not important'. The mean priority score was calculated for the resulting rated questions and ranked (online supplemental table S3) to create a list of the top 30 priority research questions.

## Workshop: determining the top 10 research priorities

We invited a subset of survey respondents and other stakeholders including NGO representatives, healthcare professionals, public members including people with lived experience of overweight to take part in a 3-hour online workshop in the winter of 2020 to determine the top 10 questions. This was a real-time, facilitator-led consultation, replacing a full-day in person event that was not possible due to local COVID-19 restrictions. The group was representative of the multidisciplinary stakeholders involved in the project; patients and members of the public, researchers, policy-makers, clinicians and relevant research funders. The workshop was held via a videoconferencing platform (Zoom), and led by external facilitators from Hopkins van Mil, a service that specialises in guiding impartial discussions to elucidate views and opinions of a diverse group of people in a safe, productive space.[12] Prior to the workshop, participants were given the resulting top 30 questions from survey 2, in addition to a list of 10 other questions from survey 1 that had been asked by more than 10 people (online supplemental table S4). The difference between the mean ranked scores in survey 2 was subtle. Workshop participants were offered the opportunity to advocate to include any of these extra 10 that they felt should be considered in the workshop to be as inclusive as possible. The workshop was divided into small groups of 4–6 people representing the range of stakeholders involved, to balance expertise and experience. Each small group was guided by a Hopkins van Mil facilitator.

The 3-hour workshop was divided into three parts with a final plenary session. Throughout the workshop participants were asked to justify their choices, and reveal the values and reasoning behind their prioritisation. Important questions were defined as those that would have the most impact if answered by research. In the first session, each group was asked to debate what they considered to be the four most and least important research questions from the 30 questions. In session 1, the highest and lowest questions were determined.

In session 2, facilitators shared a list of questions that were of medium importance, that is, not the highest or lowest priority questions determined in session 1. Facilitators asked participants to categorise these as either: (1) a priority, (2) low priority, (3) not a priority. This was determined by debate, discussion and justification of the participants' reasoning. The facilitator moved the questions around on the shared slide. The highest ranking questions from session 1 and session 2 were brought together. By the end of the second session each group had a list of top 14 questions ranked in order of importance.

The facilitators then met to combine the top 14 questions from all the small groups, this led to one list of 16 questions. In the third session of the workshop, this combined list of 16 questions was shared with the individual groups for debate. Here the groups were asked to determine and rank their final list of 10 research questions. Facilitators guided this final prioritisation stage by asking groups to focus on questions that would have the highest impact if taken forward as a research question. The groups then came together in a final plenary session and the top two questions from each group were shared with the larger group. After an amalgamation of the top two questions from each group and invariable overlap, the third and fourth questions from each group were added to produce a final list of the top 10 questions.

## RESULTS

This three-stage prioritisation project involved a diverse group of stakeholders in prioritising a list of top 10 unanswered research questions for obesity and weight-related research, which are presented here and at: https://www.phc.ox.ac.uk/research/participate/power

### Survey 1: gathering questions and identifying those unanswered

Demographics of respondents and questions gathered:

Survey 1 received 278 responses (table 1), yielding 941 original questions (figure 1online supplemental table S2). Demographic information collected during the survey indicated a diverse range of ages, ethnicities and stakeholder groups among survey respondents. Thirty-seven per cent of respondents had lived experience of obesity, and 80% were educated to degree level or above (table 1).

Identifying unanswered questions:

The 941 questions were grouped by topic. We excluded 49 (5.2%) questions as already answered, and 149, (15.8%) as out of scope (figure 1, online supplemental table S2). The remaining 743 questions were rephrased following the process above to yield 149 individual research questions (online supplemental table S3). These questions covered a range of topics (figure 2). Of the 941 submitted questions most questions concerned: prevention and intervention; mental health; illness, disease and health; and food industry, policy and environment (figure 2A). Of the 149 grouped research questions taken forward 'illness, disease and health' and 'metabolism, physiology and appetite', were the most popular categories and fewer questions concerned age of onset and duration of obesity (figure 2B).

### Survey 2: prioritising unanswered research questions

Survey 2 received 405 responses; 61% of respondents reported lived experience with obesity and 74% held an education to degree level or above (table 1). A total of 149 questions to be taken forward from survey 1 were divided into three batches of up to 50 questions, and randomly assigned to respondent's survey 2. Each question was rated in order of importance, by a mean of 115 people (SD 9.7) (online supplemental table S3).

### Workshop: determining the top 10 research priorities

We invited 64 stakeholders, 39 people confirmed their acceptance and one person dropped out on the day. Thirty-eight attendees (20 female, 18 male) were made up of 4 public members, 8 participants from related organisations, 13 researchers, 7 policy makers and 6 healthcare professionals. One person asked for the question on the role of the gut microbiome to be included from the list of 10 extra questions. At the workshop 31 questions were debated in small groups. In the first session the groups sorted questions into highest and lowest priority. In the second session, the top 14 questions were determined by all groups except one that determined their top 10 and

**Table 1** Demographic characteristics for respondents to survey 1 and survey 2

|  | Survey 1 | Survey 2 |
|---|---|---|
|  | N=278 | N=405 |
|  | n (%) | n (%) |
| Age |  |  |
| <18 | 0 (0) | 1 (0.2) |
| 18–29 | 38 (13.7) | 39 (9.6) |
| 30–39 | 47 (16.9) | 54 (13.3) |
| 40–49 | 73 (26.3) | 81 (20.0) |
| 50–59 | 69 (24.8) | 79 (19.5) |
| 60–69 | 40 (14.4) | 88 (21.2) |
| ≥70 | 8 (2.9) | 59 (14.6) |
| Prefer not to say | 3 (1.1) | 6 (1.5) |
| Gender |  |  |
| Female | 210 (75.5) | 284 (70.1) |
| Male | 61 (21.9) | 115 (28.4) |
| Non-binary | NA | 2 (0.5) |
| Other | 5 (1.8) | 1 (0.2) |
| I prefer not to say | 2 (0.7) | 3 (0.7) |
| Place of residence |  |  |
| England | 245 (88.1) | 361 (89.1) |
| Scotland | 14 (5) | 15 (3.7) |
| Wales | 8 (2.9) | 6 (1.5) |
| Northern Ireland | 4 (1.4) | 4 (1.0) |
| Not in UK | 4 (1.4) | 13 (3.2) |
| I prefer not to say | 3 (1.1) | 6 (1.5) |
| Ethnicity |  |  |
| White/white British | 236 (84.9) | 187 (85.4) |
| Mixed/multiple ethnic group | 11 (4) | 5 (2.3) |
| Asian/Asian British | 15 (5.4) | 11 (5) |
| Black/black British | 4 (1.4) | 5 (2.3) |
| Other | 3 (1.1) | 5 (2.3) |
| I prefer not to say | 9 (3.2) | 6 (2.8) |
| Education |  |  |
| School (pre-GCSE) | 4 (1.4) | 3 (1.4) |
| School (up to GCSE or equivalent) | 12 (4.3) | 14 (6.4) |
| School (A levels or equivalent) | 9 (3.2) | 11 (5.0) |
| Higher education (eg, college) | 23 (8.3) | 24 (11.0) |
| Degree level or higher | 223 (80.2) | 161 (73.5) |
| Other | 3 (1.1) | 2 (0.91) |
| I prefer not to say | 4 (1.4) | 4 (1.8) |
| Total for education question | 278 | 219* |

Continued

**Table 1** Continued

|  | Survey 1 | Survey 2 |
|---|---|---|
| Lived experience of overweight | | |
| Yes | 103 (37.1) | 248 (61.2) |
| No | 164 (59.0) | 140 (34.6) |
| Other | 11 (4.0) | 11 (2.7) |
| I prefer not to say | NA | 6 (1.5) |
| Stakeholder category† | | |
| Healthcare professional | 22 (8.6) | 36 (8.9) |
| Public health professional | 17 (6.7) | 26 (6.4) |
| Researcher (general) | 32 (12.5) | 33 (8.1) |
| Researcher (weight/ obesity research) | 49 (19.14) | 41 (10.1) |
| Work in the charity sector | 14 (5.5) | 18 (4.4) |
| Work with a group representing people with obesity | 5 (2.0) | 5 (1.2) |
| Policy-maker or commissioner of healthcare services | 1 (0.4) | 6 (1.5) |
| Professional working outside of healthcare | Not asked | 12 (3.0) |
| General interest (survey 1). | 103 (40.2) | 176 (43.5) |
| Responding in a personal capacity (survey 2) | | |
| Other or I prefer not to say | 13 (5.1) | 52 (12.8) |

*This question was not included for the first 186 respondents in survey 2.
†Stakeholder category: in survey 1 participants were able to describe themselves as belonging to more than one category. Categories selected presented. Lived experience was included in this section.
GCSE, General Certificate Secondary Education.

another that grouped questions as high medium and low priority. The facilitators combined the lists from all the groups into a list of 16 as many of the top 14 from each group overlapped. In the third session participants ranked the 16 questions into a top 10 list. The top two from all these lists was shared at a final plenary session. Consolidation of the top two questions and the questions ranked third and fourth resulted in 11 unique research questions by the end of the workshop. On analysis of recordings of each group's discussion, multiple participants noted that two questions in the 11 that were similar in meaning. We, therefore, combined these two questions (concerning food choice, appetite and the brain's control of food intake) post hoc to produce a final list of

the top 10 research questions (box 1). There was consistency between the top questions in this final list produced from the workshop, and popular questions submitted in the surveys as indicated by number of people asking each question (online supplemental table S3). Five of the final top 10 questions were among the 10 most frequently submitted questions in survey 1. Seven of the final questions were in the top 10 from survey 2, ranked by mean score. The final list of the top 10 priorities are not listed in order of priority (box 1).

## DISCUSSION
### Summary of findings
Our priority setting exercise identified the top 10 research questions that stakeholders in the field, and those with an interest overweight and obesity, believe to be the key priorities to advance obesity and weight-related research. In two online surveys and a workshop, we collated nearly 1000 questions, and guided participants in a structured and systematic prioritisation process to reach the final list of 10 (box 1). These questions cover a wide spectrum of areas, and if answered by research, would generate knowledge applicable for individuals, healthcare, public health and policy.

### Strengths and limitations
The main strength of this project was the successful collaboration between a diverse range of stakeholder groups, though it was not without its challenges. Our stakeholders included patients, members of the public, clinicians, charities, researchers and policy-makers connected to the field. Identifying the most appropriate group for a condition where a third of the adult population are clinically obese and more than 60% are overweight, while at the same time, considering how to prevent the condition developing which is relevant to the whole population, resulted in the inclusion of both patients and members of the public. Members of the public naturally included some people without lived experience of overweight or obesity, though it seems unlikely that they would not be aware of family members, and friends who are affected, and they may become affected themselves, justifying their inclusion in this prioritisation process. To have found consistency in the questions being posed throughout the entire process by a variety of individuals bringing different experience and expertise to a common area of focus, supports the validity of the resulting top 10 questions. The majority of survey respondents and workshop participants appear to be highly educated. Nonetheless there was evidence of an awareness of the need for interventions to help reduce inequalities and the top 10 priorities include questions on social determinants of health like low-socioeconomic status and cultural factors. We commissioned third-party, impartial facilitators to guide the workshop without input from the research team, so as to not inadvertently sway the prioritisation of questions being considered in each session. Additionally, the

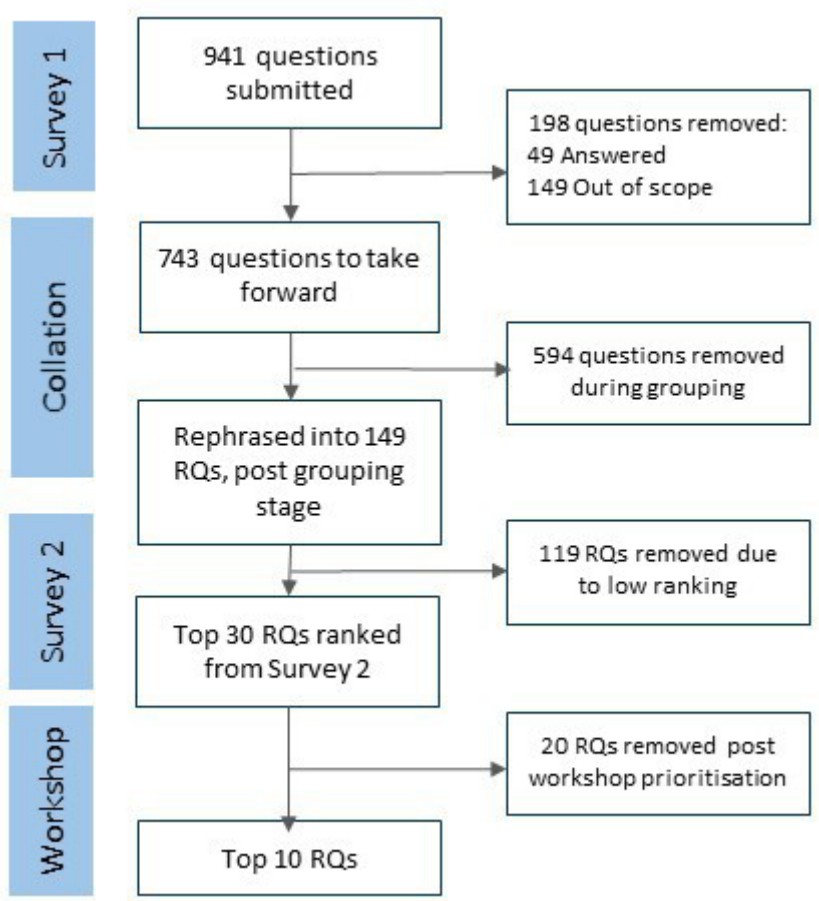

**Figure 1** Flow diagram for the priority research questions. RQ, Research questions.

number of questions submitted and finally categorised is in line with similar priority setting exercises in health research, using an analogous process set out by the JLA, with a comparable number of stakeholders involved.[13 14]

There are limitations that we identified and strived to address throughout the stages of the process. To begin with survey 1, we deemed that 5.2% of all questions submitted were already answered by empirical evidence. We assessed this through a thorough search of

the literature to identify systematic reviews, clinical and public health guidelines and high-quality primary studies in the form of RCTs. Although this necessitated some subjective judgement, we ensured that all decisions were made in duplicate, and discrepancies were resolved by a third researcher. Our confidence in the categorisation of answered versus unanswered questions is strengthened by consensus among stakeholders involved, some of whom were researchers with expertise in the question

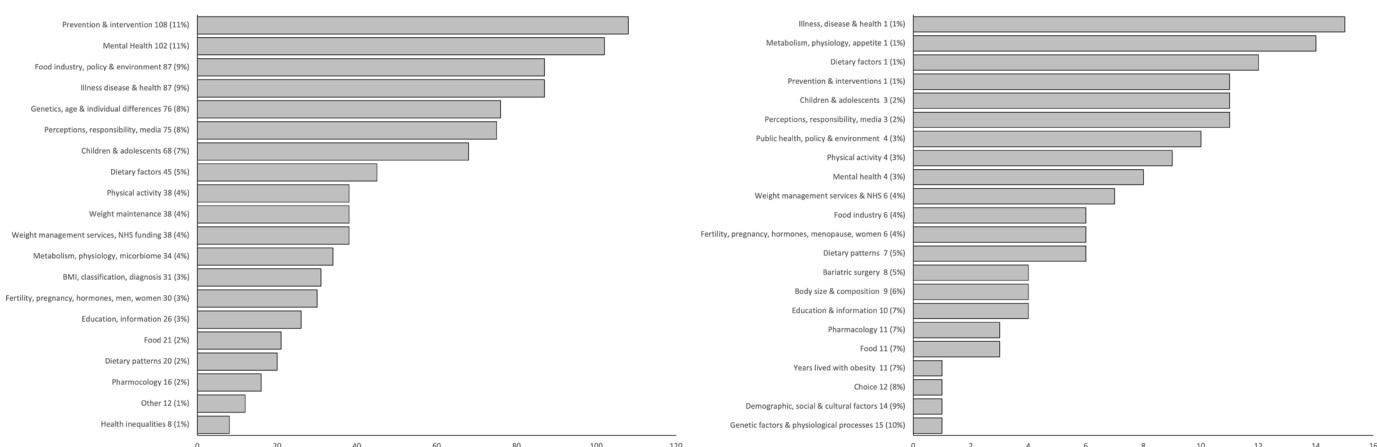

**Figure 2** (A) 941 submitted questions by topic; (B) 149 research questions grouped by topic. BMI, body mass index; NHS, National Health Service.

## Box 1 Final top 10 priority research questions*

What are the most effective methods for weight maintenance following weight loss? What are the effective components of treatments/programmes incorporating a behavioural element? How many and in what combination are most effective? What is the optimal duration of these programmes?

What are the most effective methods for weight loss? What are the effective components of treatments/programmes incorporating a behavioural element? How many and in what combination are most effective? What is the optimal duration of these programmes?

What is the most effective and cost-effective mix of population/public health and individual interventions to tackle obesity?

Do interventions that target the 'obesogenic environment', such as community interventions, urban planning, placement of fast-food outlets or workplace polices, affect population mean weight and do these effects differ by baseline weight status (underweight, healthy weight, overweight, obesity)? Which interventions are most effective at reaching low socioeconomic groups?

Do interventions (eg, nutrition education and physical activity) in preschool, primary school and secondary school reduce children's risk of unhealthy weight gain and, if so, how do they act? Does the effect of such interventions differ by social and cultural groups?

What changes in supermarkets or the wider food industry are effective in promoting healthier diets? Does changing labelling and/or packaging on foods affect purchasing, consumption and body weight?

What is the cost and affordability of a healthy balanced diet? How can we make healthier foods more affordable? How can we improve access to healthy diets for social and cultural groups, such as people in poverty, people in inner cities, or young and older people?

How do demographic, social and cultural factors (eg, age, socioeconomic status, lifestyle, environment, psychosocial functioning) affect weight status, weight gain and regional fat distribution? What are the mechanisms involved? Does the effectiveness of weight loss methods depend on social and cultural background and, if so, can the effects be made more equitable? Are weight loss methods tailored to people's background more effective for weight loss and weight maintenance than general methods?

How accurate are existing models of health consequences of excess weight and the impact of weight loss? Which assumptions are critical in determining the long-term effectiveness and cost effectiveness of weight loss interventions? What do these models predict is the impact of weight loss interventions on health and disease incidence and the cost-effectiveness of such interventions? What is the impact of weight regain on the incidence of disease and cost-effectiveness of weight loss interventions?

What are the drivers of food choice, appetite and intake and do variations in these drives explain who develops obesity and who does not? How does the brain control food intake and can we use these mechanism to aid weight loss? What are the brain responses (neural correlates) in response to food during weight loss and following weight regain?

*Footnote to box 1 *These questions are in no particular order, that is, not in order of importance.

---

areas being considered. That 5.2% of the submitted questions were considered answered indicates that research may not being adequately communicated in these areas. This could be addressed by improved or targeted communication.

In survey 2, we asked participants to rate questions on a scale of 1–10, but found that participants were disinclined to use the full range. Many questions had means between 6 and 8 meaning that differences in the scoring were subtle. Future work could consider using a condensed scale to perhaps mirror ratings that people are more familiar with (eg, 5-point ratings seen in 5-star reviews or 4-point grading of evidence[15] however unless people used the top and the bottom of the scale a condensed scale could lead to questions being rated as even more similar. In regards to the workshop, while facilitators had standardised methods for the structure of the small group discussion, one group did not rank their questions during the workshop, and instead batched them as high, medium and low priority. This made no difference to the outcome, as the group's top three high-priority questions were included in the final priority list across all groups.

An additional limitation of the workshop was the shift to a virtual vs in-person meeting due to local COVID-19 restrictions, which limited the length of the discussions. On analysis of the recordings from each group's consultations, it was clear that the virtual setting maintained a clean discussion where moderators were clearly able to garner input from each participant without anyone talking over-another, as may have been the case in an in-person discussion. It was possible to rank the top 10 in the small groups in the workshop, however, it was harder to achieve this with 38 participants in the plenary sessions so we did not seek to order the final 10 priorities. Lastly, obesity is a worldwide problem calling for a global research response, but we only involved UK-based stakeholders. It is likely that the process identified questions that are generalisable outside of the UK, but it is unlikely that this process fully captured priorities that may be relevant to low-income countries or countries with a low prevalence of overweight and obesity.

### Implications for research and policy

The main implication of this work is for the top 10 POWeR to be considered by funding bodies concerned with advancing the field. Similar priority setting work in other areas of health research have resulted in research calls that reflect priorities identified by stakeholders.[16] Along with the top 10 questions, this project identified a further 139 unanswered questions that may also serve as a resource for researchers trying to match gaps in the evidence with perceived needs.

We make two recommendations for future priority setting exercises in this area based on what we learnt from the process. The first would be to limit the focus to a prespecified area in the field. The breadth of topic areas produced by the large number of stakeholders engaged was onerous to process, and resulted in high level areas for research. Future exercises may wish to restrict their scope to a certain area of research, such as treatment for people living with obesity, or population-wide prevention strategies to allow more granular questions to emerge. The second would be to incorporate work

to boost awareness of the existing research evidence for common questions that were submitted, but deemed to be answered. Questions that were commonly submitted, but already answered and therefore excluded from progressing through the prioritisation process are telling of a discrepancy between published research and knowledge dissemination.

## CONCLUSION

We have identified 10 priorities that cover: the role of the obesogenic environment; effective weight loss and maintenance; prevention in children; effective prevention and treatment policies; the role of the food industry; access to and affordability of healthy diets; the sociocultural factors associated with weight; the biology of appetite and food intake and long-term health modelling. Research funders may want to prioritise these questions when considering research proposals, or commissioning programmes of research to answer these key questions.

**Acknowledgements** We would like to thank Nicola Lindson and Jamie Hartmann Boyce for help and guidance and David Dyson, Gavin Hubbard, Alice Crouch and Dan Richard-Doran for support with the surveys and webpages, Michaela Noriek for help with public engagement, Goher Ayman for helpful advice We also want to thank everyone who worked on the POWER project including all the members of the health behaviours team who helped us with questions from survey 1. Elizabeth Morris, Carmen Piernas-Sanchez, Dimitrios Koutoukidis, Tanisha Spratt, Anne Ferrey, and Jenny Brooks. We would like to thank all the people who took the surveys and provided us with their original questions, everyone who ranked the questions and to all the workshop participants and the many organisations that distributed the surveys on our behalf. We are grateful to the workshop participants for giving us their time. We would like to thank Hopkins Van Mil for conducting the workshop.

**Contributors** NMA, PA and SAJ conceived the concept. ARB, LG and NMA were responsible for collecting and collating data with guidance from PS and BC. ARB, LG, AH and NMA produced an initial draft of the manuscript, PA, SAJ, PS and BC provided comments and edited the original draft. All authors reviewed and approved the final submitted version of the manuscript. NMA is responsible for the overall content as the guarantor and accepts full responsibility for the work and/or the conduct of the study, had access to the data and controlled the decision to publish.

**Funding** The study was funded by the National Institute for Health Research (NIHR) Oxford and Thames Valley Applied Research Collaboration. NMA, PA and SAJ are supported by the NIHR Oxford Biomedical Research Centre. PA and SAJ are NIHR senior investigators.

**Disclaimer** The funders had no role in study design, data collection, data analysis, data interpretation, or writing of the report. The views are those expressed by the authors and not necessarily those of the NHS, NIHR, or Department of Health.

**Competing interests** NA, PA, and SAJ led an investigator-initiated study funded by Cambridge Weight Plan. PA has spoken at two symposia organised by the Royal College of General Practitioners that were funded by Novo Nordisk. None of these activities led to personal payment. ARB, LG, AH, PS and BC have no interests to declare.

**Patient and public involvement** Patients and/or the public were involved in the design, or conduct, or reporting, or dissemination plans of this research. Refer to the Methods section for further details.

**Patient consent for publication** Not applicable.

**Ethics approval** The study was approved by the University of Oxford Medical Sciences Inter Divisional Research Ethics Committee (Ref: R6721/RE003). Participants gave informed consent to participate in the study before taking part.

**Provenance and peer review** Not commissioned; externally peer reviewed.

**Data availability statement** Data are available on reasonable request. Requests can be made for the deidentified participant level data collected during this study

from the Nuffield Department of Primary Care hosted Datasets Independent Scientific Committee (PrimDISC): primdisc@phc.ox.ac.uk on approval of a protocol, statistical analysis plan and the signing of a suitable data sharing agreement.

**ORCID iDs**
Ailsa R Butler http://orcid.org/0000-0002-8577-6574
Nerys M Astbury http://orcid.org/0000-0001-9301-7458
Paul Aveyard http://orcid.org/0000-0002-1802-4217

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
