## [Reviewer comments · BMJ Open]

ARTICLE DETAILS

TITLE (PROVISIONAL)	Setting the top 10 priorities for obesity and weight-related research (POWeR): a stakeholder priority setting process
AUTHORS	Butler, Ailsa R; Astbury, Nerys; Goddard, Lucy; Hajizadeh, Anisa; Seeber, Philippa; Crawley, Bruce; Aveyard, Paul; Jebb, Susan

VERSION 1 – REVIEW

REVIEWER	Dean, Caitlin Academisch Medisch Centrum, Obstetrics and Gynecology
REVIEW RETURNED	27-Oct-2021

GENERAL COMMENTS	This is a priority setting project or obesity and weight related questions, loosely following the James Lind Alliance (JLA) methodology. It is clear the authors have followed an number of the prescribed steps, however the detail needs to be given in the manuscript so that readers can understand how the priorities were achieved and to assess if they are truly representative of what is important to those they seek to benefit. Clarity of the decision making process would strengthen this manuscript and acknowledgement of where methods have deviated from the JLA process. Further specific suggestions and questions are below, I hope they are useful to the authors. Introduction You state that you followed the James Lind Method for this. Did you have a JLA advisor overseeing the project and if so why are they not a co-author? If you followed this method you need to justify your deviations from their method. You need to state how obesity is being defined and also what “weight related research” specifically refers to. Page 5 Line 32 – “In the field of obesity and other questions related to excess weight, there is no obvious patient constituency as we are all at risk of developing overweight and the perspective of policy-makers need to be included as well as clinicians” – While I agree that policy makers need to be included and there is good justification for this, the idea that there is no obvious patient constituency seems not in the spirit of a James Lind Alliance project. Just because anyone can become overweight (which isn’t actually true anyway) doesn’t mean that anyone can represent that particular group of patients. The patient constituency for this project are people who are currently obese and particularly those with weight related disease or those who have previously been obese. Please provide a more careful discussion of this and greater justification for including public without lived experience of obesity (it may be a partner, parent or child of theirs who is obese
---

	but they have to have some understanding of it), particularly given the stigmatisation of obesity in society and the risk of bias introduced by people without genuine lived experience of it. Page 6 line 22 – please be more explicit in your description of patients who gave input. Presumably just because someone is obese it doesn't make them a patient, but they have lived experience of obesity, whereas someone with obesity related disease would be considered a patient. Do the public members PS and BC have lived experience of obesity? Did people with lived experience give feedback on materials such as the wording of the survey? Some additional context of why this research is needed, how obesity and weight related research has perhaps been misdirected in the past, or not to patient benefit and how these priorities will benefit patients and the healthcare professionals caring for them would strengthen the justification for this project and increase the chance of the priorities being addressed by researchers and funders. Stage 1.  • Did you have a protocol? • Did you convene a steering group – who was on it and what were their specialities or interests and what organisations were represented? • What was the scope more precisely? Was this UK specific and was it limited to obesity in adults? • Did you pilot the survey and it's wording? • A table with which organisations distributed it and an indication of their reach would be useful. • How were harder to reach participants attempted to be included... were there any paper surveys? Did you specifically try to include different cultures. • How was the literature searched – did you have a pre agreed protocol for this process? Was there a time limit on systematic reviews to be considered up to date, ie. 3 or 10 years? • Did you not form the indicative questions prior to evidence checking as per the JLA method? • What did you intend to do with unknown knowns? Stage 2.  • How was your online survey distributed at this stage? • What software was used for the survey? • Did participants from the first stage have the opportunity to take part in the second stage? • You have not defined RQ. • You have not specified the direction of importance of the 1-10 - was 1 considered most important or least important? Stage 3. This section gives far more detail than necessary, particularly given the paucity of detail in stages 1 and 2. I would suggest you cut some of this and use the words to give greater detail in the previous sections. Results Stage 1. How long was the survey open for? I would suggest reworking supplementary table 1 to include:
--	--

	 • A column indicating either that it considered to be answered (ideally with a reference to the answer), was removed as out of scope/not relevant (with a code for their reason). • A column for those included you put the indicative question it was incorporated into, 1-149. • A column indicating the category of participant who asked it (public/researcher/healthcare professional etc) This would demonstrate that every raw question was dealt with and either included in an indicative question or excluded. You could then get rid of Supplementary table 2. It would also enable us to see if it is mostly patient/public questions which were excluded or not. Some of the excluded questions seem like good questions and it isn't clear why you have excluded them. For example: "Are there safe drugs available to treat the condition?" Why should this not be a research question. A lot of the ones you state are not research questions it is clear what the person is asking and that it could be merged into an indicative question, and if you're asking the general public they are going to ask question in a lay format. I think you need greater justification for your exclusions, ie. Were they discussed and agreed within the steering group? I appreciate that the details of participants are in table 1 but a brief description of key characteristics, particularly the category of patient/researcher/healthcare professional and the overlap of lived experience would be useful in the text.  • I can't see that figure 1 is referred to before figure 2 • How were the categories in figure 2 identified? • Greater detail on the "answered questions" would be useful. Search terms, results and screening process etc. Stage 2  • Page 9 line 31 – I think you mean Table 1? • When/how long was the survey open for? Stage 3.  • Please provide details of the participant representation, ie. Lived experience/ researcher/what professions they represented • Table 2, I would suggest adding numbers 1-10 in a column to the left Discussion I think some discussion of your work in the context of other JLA priority setting projects is needed. How does your uptake compare with other projects? Your top ten questions mention people from low socio-economic groups, people living in poverty and inner cities, children and different social and cultural groups, yet you do not appear to have representation of these in your participation. For example, the vast majority of participants were educated to degree level or above. This needs to be contextualised in your discussion. You have deviated from the JLA method in a number of ways, in particular in your stakeholder representation by having a lot of researchers involved and public without lived experience who could bring all sorts of misunderstanding and stigma with them. This needs reflection in the discussion and contextualised with other priority setting projects and the wider field of obesity literature.
--	--

REVIEWER	Brütt, Anna Levke University of Oldenburg School of Medicine and Health Sciences, Department of Health Services Research
REVIEW RETURNED	08-Nov-2021

GENERAL COMMENTS	Summary This manuscript reports on an important subject and the study was carried out with very high effort. The reporting of the methods and results needs to be improved as there were many open questions left. I would further recommend checking the language and grammar. Detailed comments (number/page/line): 1 3 25 Does most important mean top 10? If so, please state this here already. 2 3 11-25 Use already the terms survey 1, 2 and 3 in the method part for the different surveys conducted to avoid misunderstanding. 3 5 9-11 The sentence needs rewriting. 4 5 7-20 You might write a little bit more on the importance of the problem of obesity to the health care system, like what kind of major diseases are related to obesity, what are the consequences for the health care system,... . Also, make clearer what might be the reasons, why existing evidence did not lead to a reduction in obesity so far – is this because the existing research is not helpful for decision-makers to derive political action from it? Did the research not address the most relevant questions so far (because certain people like patients are caregivers where not involved in setting the research agenda)? Or are there political/economic reasons, why existing evidence could not have been transferred to practice yet? 5 5 32-41 Make clearer, why also the viewpoints of politicians need to be considered in setting the research agenda for obesity (or overall public health topics) or respectively, why they do not need to be considered in classical JLA approaches. Also, I do not fully understand the section “there is no obvious patient constituency as we are all at risk of developing overweight” – This argument might also be the case for several other diseases not exclusively caused by genetic reasons (e.g., we are also all at risk to develop a depression or having a heart attack). I do think that for obesity or overweight related health problems there is an obvious patient constituency like for most other diseases as well or I do not fully understand what you are trying to say here. 6 5 39-41 State, what this modification to the JLA process includes. 7 5,6 52-57, 3-6 As you refer to the JLA process several times, it would be helpful for the readers to provide a short description of the general/classic JLA process and state, what you have adapted/modified. When you write “but including members of...”, it sounds like in the general JLA process these stakeholder groups are normally not involved but this is not true for some of these stakeholders. 8 7 6-8 How have you assessed the questions according to the category “answerable by empirical research”. Also, was this the first step and afterwards you categorised the questions further into “already answered” and “unanswered”? If so, writing these three categories in one sentence might be misleading as it can also be understood that you have divided the questions according to these three categories.
---

	9 7 10-13 Did you also have a time limit, like no review older than 3 years (like the JLA suggests it). To which guidelines for assessing the evidence do you refer here? Please also state how you have searched for evidence (e.g., which data bases?) 10 7 12-17 Please describe the grouping and formulation process of the re-search questions more detailed. Did you had a coding system for grouping the research questions? Have you used some guide-lines/frameworks to formulate the research questions? 11 7 30-34 Describe, how you batched the questions and how many ques-tions were left. Also, you use the abbreviation RQ only once here without explanation. I would recommend to write research ques-tion instead. Please also state, which online tool you have used for the ques-tionnaire. 12 7 40-45 Say, how many people you have invited and which online-tool you used for the virtual workshop. 13 7 45-49 Why have you decided to include the extra 10 questions here as well? Please state this clearer. 14 7,8 section "Stage 3" I would recommend to bring more structure into the de-scription of stage three. First state, what was the aim of this process, then who was involved, how many and what the participants needed to prepare. Afterwards, describe the process used for pri-ority setting (dividing participants into smaller groups, discussion in groups and so on). 15 8 53-57 How was the top 10 produced from the fourth highest ranked questions? Can you provide more detail on how the top 10 were produced and how all participants agreed on the final list? (dis-cussion or voting like an example) 16 9 14 write the number instead of "Seven-hundred and forty-three" 17 9 19-21 You could provide more information here on the final included questions in stage one. At least indicate the mentioned topic are-as. You could also indicate if some topics areas were more repre-sented, i.e. included more questions than others. 18 9 27 write numbers instead of "Four-hundred and five" 19 9 26-31 I do not fully understand what you want to say here. Did batching mean that not all persons got the same questionnaire, i.e. the same questions for rating? In the method part I understood that you further combined questions (summarizing similar questions to reduce the overall number of questions) and all participants than got the same questions. 20 9 24-32 Provide more information on the rating of the questions (which questions were high ranked and included in stage 3? Are the high ranked topic areas somewhat similar?, Were the participants unanimous in their rating? (provide consensus values),...) 21 9 36 Write the number instead of "Sixty-four" 22 9 50 I get a little bit confused here – you write top 10 questions here but wasn't the aim to rank the top 14 for each group? Why did the group not rank the questions? Due to lack of time? 23 10 15-22 I found this paragraph difficult to understand. You write here for the first time, that you also ranked the questions for survey 1 according to the number of people asking the question. This might confuse the reader as you write in the method part, that you used stage one to gather uncertainties first and used the collected questions for ranking in stage 2 and 3 (no rank-ing in stage 1). Per-haps you can already mention this earlier in the method part that you will com-pare the top ranked questions between each stage.
--	---

	24 11 13-17 I would say these sentences are more the reporting of results than the dis-cussion of them. Provide more information on the last sentence, which topic areas were covered by the top 10 ques-tions. Which public health problems/policy problems are ad-dressed by the ques-tions? 25 11 32-36 It sounds here like you used the three methods independently to identify and prioritise the top research questions. But it was a process where you reduced the questions in each stage to the final top 10. I would say that it is therefore not really possible to say that the rank-ing of the questions was not affected by the methods in each stage as the stages were not inde-pendent from each other. 26 11 38-45 Please make clearer in the method part, which guidelines you have used and how you have searched for evidence. When you had objective guidelines and a transparent way to exclude re-search questions that were already answered I would not say that this process it to subjective so you might formulate this a little bit more positive here. 27 11 discus-sion section I am missing here a little bit more discussion related to the re-search that has been done so far on the issue of obesity com-pared to the research questions you have identified. Can you for example say, that your research questions represent other inter-ests than previous research? 28 12 45-54 I would mention these topic areas already in the result section. Lisa-Ann Baumann helped in completing the review.
--	--

VERSION 1 – AUTHOR RESPONSE

Thank you for taking the time to review our paper, we have amended the manuscript in response to the reviewers comments, as detailed point by point below, and these changes have been highlighted. We have also taken the opportunity to improve flow and clarity of the submission:

Introduction

1. You state that you followed the James Lind Method for this. Did you have a JLA advisor overseeing the project and if so why are they not a co-author? If you followed this method you need to justify your deviations from their method.

JLA were contacted to be involved, however they were not interested in being involved with a priority setting exercise that deviates in any way from their usual approach.

Our main deviation from the JLA approach is that the input was not limited to patients with lived experiences of the condition and clinicians who treat it, and we have made this clear.

2. You need to state how obesity is being defined and also what “weight related research” specifically refers to.

We have now defined obesity, and also included a description of what we mean by weight related research:

P5

“The detrimental effects of excess weight are not restricted to those who meet the BMI threshold of obesity as the increased morbidity is seen in people with any degree of excess adiposity7.

Accordingly, strategies to prevent obesity or excess weight or adiposity are needed, defined here as obesity and weight-related research.”

3. Page 5 Line 32 – “In the field of obesity and other questions related to excess weight, there is no obvious patient constituency as we are all at risk of developing overweight and the perspective of policy-makers need to be included as well as clinicians” – While I agree that policy makers need to be included and there is good justification for this, the idea that there is no obvious patient constituency seems not in the spirit of a James Lind Alliance project. Just because anyone can become overweight (which isn’t actually true anyway) doesn’t mean that anyone can represent that particular group of patients. The patient constituency for this project are people who are currently obese and particularly those with weight related disease or those who have previously been obese. Please provide a more careful discussion of this and greater justification for including public without lived experience of obesity (it may be a partner, parent or child of theirs who is obese but they have to have some understanding of it), particularly given the stigmatisation of obesity in society and the risk of bias introduced by people without genuine lived experience of it.

We have now removed this sentence and replaced it with a statement that a wider range of stakeholders were included (and is how our approach deviates from the JLA approach):

P5 end - P6 lines 1-11

“Policies to prevent obesity typically affect the whole of society, for example fiscal policies or policies restricting the promotion or selling of some goods. Likewise, providing treatment for obesity as part of publicly funded healthcare is contested, and thus questions about research in this area seem to call for a much wider group of stakeholders than patients, carers, and clinicians..”

4. Page 6 line 22 – please be more explicit in your description of patients who gave input. Presumably just because someone is obese it doesn’t make them a patient, but they have lived experience of obesity, whereas someone with obesity related disease would be considered a patient. Do the public members PS and BC have lived experience of obesity? Did people with lived experience give feedback on materials such as the wording of the survey?

We have re-written this section to make this clearer. We piloted both surveys and the wording and layout of our webpage with members of the public.

P 6 “We involved two members of the public (BC and PS) with lived experience of overweight in all stages of the project, from conception and design of the study, to its conduct, data collection and analysis. Our wider public involvement (surveys and workshop) incorporated members of the public with and without lived experience of being overweight and patients, defined as people with lived experience of being overweight and experience of receiving clinical treatment for overweight, obesity, or an associated condition. Members of the public were involved in all stages of the work alongside and as equal partners with other stakeholders.”

Piloting:

P 8 – survey 1

“The survey was administered online using JISC Online Survey and was piloted prior to being launched.”

P10, survey2:

“The second online survey was piloted with members of the public and colleagues.”

5. Some additional context of why this research is needed, how obesity and weight related research has perhaps been misdirected in the past, or not to patient benefit and how these priorities will benefit

patients and the healthcare professionals caring for them would strengthen the justification for this project and increase the chance of the priorities being addressed by researchers and funders.

We have now added additional context to explain why this research is needed now:

Background p4:

“Presently, the research agenda is mainly driven by the interests and concerns of researchers, or research commissioners. A more transparent, systematic, and collaborative approach involving multiple stakeholders to identify research priorities could accelerate progress..... “

6. Stage 1.

- Did you have a protocol?

Yes we developed a protocol to guide the prioritisation process for the online survey and workshop prioritisation exercises and have now specified that we followed a protocol for this project:

P7

“The protocol for the project was approved by the University of Oxford Medical Sciences Inter-Divisional Research Ethics Committee (Ref: R6721/RE003).”

- Did you convene a steering group – who was on it and what were their specialities or interests and what organisations were represented?

There was a study management group consisting of the investigators and PPI representatives, with lived experience of overweight, which was convened during the study to guide the process, we have added further information into the submission to clarify this:

P7

“There was a study management group of investigators and PPI representatives that met regularly.”

- What was the scope more precisely? Was this UK specific and was it limited to obesity in adults?

We have now added a section to clarify the scope, see below.

P7

“The scope was limited to research questions on the aetiology, consequences, prevention or treatment of overweight and obesity in both adults and children , and did not include questions about whether currently evidenced interventions or policies should be implemented.”

- Did you pilot the survey and it’s wording?

Please see response to comment #4

- A table with which organisations distributed it and an indication of their reach would be useful.

We have included a table with this information in the supplementary materials

- How were harder to reach participants attempted to be included... were there any paper surveys? Did you specifically try to include different cultures.

We did make paper surveys available,. We have now added further detail on this:

P8

“We made physical copies of the survey, and versions with a large font size readily available upon request,…”

- How was the literature searched – did you have a pre agreed protocol for this process? Was there a time limit on systematic reviews to be considered up to date, ie. 3 or 10 years?

We had a pre-specified protocol for literature searching which included keyword and MeSH terms, and we have added further detail of this:

P9

“We then searched the literature using keywords and MeSH terms informed by the questions, to determine if these were areas that were already adequately addressed in the scientific literature. Questions were deemed ‘answered’ if there was satisfactory evidence....”

- Did you not form the indicative questions prior to evidence checking as per the JLA method?

Yes. See section below:

P9

“Survey 1 questions were grouped by topic area and rephrased to form answerable research questions. We used a multi-level coding system to categorise questions into overarching categories that were iteratively deduced throughout the grouping. “

- What did you intend to do with unknown knowns?

We excluded the 49 (5.2%) questions deemed to be already answered, kept a record and made these available in our supplementary material. We note and debate the importance of these ‘unknown knowns’ in the discussion section of our paper.

We note in the discussion, P 17:

“That 5.2% of the submitted questions were considered answered indicates that research may not being adequately communicated in these areas. This could be addressed by improved or targeted communication.”

7. Stage 2.

- How was your online survey distributed at this stage?

We distributed Stage 2 survey to all those who completed Stage 1, and also via a raft of organisations as per stage 1. We have now added this detail in to the methods section and included the list of organizations in Supplementary Table 2

P10

“The survey was administered via RedCap, and sent to those who participated in the first survey, as well as to the organisations approached to share survey 1.”

- What software was used for the survey?

See above.

- Did participants from the first stage have the opportunity to take part in the second stage?
Yes, please see our clarification above. In survey 1 participants provided their email addresses if they wanted to be involved in survey 2.

- You have not defined RQ.

We now put this in full as 'research question' throughout the text.

- You have not specified the direction of importance of the 1-10 - was 1 considered most important or least important?

We have now added more detail to clarify direction of importance.

P10

"Respondents were asked to rate each question on a scale of 1-10 with 10 representing 'very important' and 1 representing 'not important'."

8. Stage 3.

This section gives far more detail than necessary, particularly given the paucity of detail in stages 1 and 2. I would suggest you cut some of this and use the words to give greater detail in the previous sections.

Thank you for this useful suggestion, we have taken this comment on board and cut detail in this section, and added more details to the Stage 1 and Stage 2 methods.

9. Results Stage 1.

How long was the survey open for?

We have added this detail into the methods section

P9, survey 1

".....and open for responses for 37 days between January 15th and February 21st, 2020."

P10

"The second survey remained open for 30 days between August 6th and September 14th, 2020."

I would suggest reworking supplementary table 1 to include:

- A column indicating either that it considered to be answered (ideally with a reference to the answer), was removed as out of scope/not relevant (with a code for their reason).

- A column for those included you put the indicative question it was incorporated into, 1-149.

- A column indicating the category of participant who asked it (public/researcher/healthcare professional etc)

This would demonstrate that every raw question was dealt with and either included in an indicative question or excluded. You could then get rid of Supplementary table 2. It would also enable us to see if it is mostly patient/public questions which were excluded or not.

Thank you for your comment, unfortunately, since we did not consider if the original submitted questions were answered or unanswered, we're unable to incorporate this into the Supplementary Tables, however, we have now re-formatted Supplementary Tables 2 and 3 to incorporate some of the information requested.

Some of the excluded questions seem like good questions and it isn't clear why you have excluded them. For example: "Are there safe drugs available to treat the condition?" Why should this not be a research question. A lot of the ones you state are not research questions it is clear what the person is asking and that it could be merged into an indicative question, and if you're asking the general public they are going to ask question in a lay format. I think you need greater justification for your exclusions, ie. Were they discussed and agreed within the steering group?

We very much tried to group the submitted questions or comments where we could sense the question behind the comment with similar questions in order not to lose these questions. These were then merged into the rephrased research questions.

All the submitted questions and grouped rephrased questions and excluded questions were discussed with the study management group including our PPI members.

I appreciate that the details of participants are in table 1 but a brief description of key characteristics, particularly the category of patient/researcher/healthcare professional and the overlap of lived experience would be useful in the text.

Thank you for this comment- we have now added a paragraph to narratively describe the participants. Full details are presented in Table 1.

P 12-13 Survey 1:

"Demographic information collected during the survey indicated a diverse range of ages, ethnicities, and stakeholder groups among survey respondents. 37% of respondents had lived experience of obesity, and 80% were educated to degree level or above (Table 1)."

P15 Survey 2:

"Survey 2 received 405 responses; 61% of respondents reported lived experience with obesity, and 74% held an education to degree level or above (Table 1)."

- I can't see that figure 1 is referred to before figure 2

Thank you we now refer to Figure 1 before Figure 2 on P 13.

- How were the categories in figure 2 identified?

We have added more detail in the text to describe this process.

P9

"We used a multi-level coding system to categorise questions into overarching categories that were iteratively deduced throughout the grouping. For example, the submitted question 'which diets work' fell into a macro category, 'treatment' and was then further filtered into the sub-category 'behavioural' over 'pharmaceutical'"

- Greater detail on the "answered questions" would be useful. Search terms, results and screening process etc.

We have included further detail in the methods section on how we searched the literature for answered questions:

See response to Reviewer 1 comment 6

Stage 2

- Page 9 line 31 – I think you mean Table 1?

Thank you. Table 1 - demographic characteristics.

- When/how long was the survey open for?

We have now added this information into the methods section

P9, survey 1

“.....and open for responses for 37 days between January 15th and February 21st, 2020.”

P10

“The second survey remained open for 30 days between August 6th and September 14th, 2020.”

Stage 3.

- Please provide details of the participant representation, ie. Lived experience/ researcher/what professions they represented

38 participants attended the workshop, these were made up of:

We have now included this information in the methods section:

P 10

“a subset of survey respondents and other stakeholders including NGO representatives, healthcare professionals, public members including people with lived experience of overweight”

P14

“The 38 attendees (20 female, 18 male) were made up of 4 public members, 8 participants from related organisations, 13 researchers, 7 policy makers and 6 healthcare professionals.”

- Table 2, I would suggest adding numbers 1-10 in a column to the left

We have not used numbers for the questions, as although the questions were identified as the top 10 priorities, they were not in any specific order, and providing a number would indicate that lower number questions are a higher priority than higher numbered ones. We have added a footnote to the table to clarify this.

Discussion

I think some discussion of your work in the context of other JLA priority setting projects is needed. How does your uptake compare with other projects?

We have now added some additional information comparing our response rate to other JLA priority setting partnerships:

P19

“...the number of questions submitted and finally categorised is in line with similar priority setting exercises in health research, using an analogous process set out by the James Lind Alliance, with a comparable number of stakeholders involved 14,15.”

Your top ten questions mention people from low socio-economic groups, people living in poverty and inner cities, children and different social and cultural groups, yet you do not appear to have

representation of these in your participation. For example, the vast majority of participants were educated to degree level or above. This needs to be contextualised in your discussion.

We have included an additional paragraph in the discussion to acknowledge that the top 10 priorities include research relevant to people from low SES, children and different social and cultural groups, not necessarily represented in the virtual workshop:

P19

“The majority of survey respondents and workshop participants appear to be highly educated. Nonetheless there was evidence of an awareness of the need for interventions to help reduce inequalities and the top 10 priorities include questions on social determinants of health like low-socioeconomic status and cultural factors.”

You have deviated from the JLA method in a number of ways, in particular in your stakeholder representation by having a lot of researchers involved and public without lived experience who could bring all sorts of misunderstanding and stigma with them. This needs reflection in the discussion and contextualised with other priority setting projects and the wider field of obesity literature.

We have included a sentence to acknowledge that although including a wide range of stakeholders broadens the scope, by not limiting respondents to people with a lived experience of obesity this may bring some mis-understanding to the condition.

Reviewer 2

Summary

This manuscript reports on an important subject and the study was carried out with very high effort. The reporting of the methods and results needs to be improved as there were many open questions left. I would further recommend checking the language and grammar.

Thank you for your comments- we have made significant changes to the methods section as detailed below:

Detailed comments (number/page/line):

1. 3 25 Does most important mean top 10? If so, please state this here already.

We have now re-written this and state

P 2 “...a final list of 10 priorities”

2. 3 11-25 Use already the terms survey 1, 2 and 3 in the method part for the different surveys conducted to avoid misunderstanding.

We have now defined each stage within the abstract. Survey 1, survey 2 and a workshop. We hope this is clear.

3. 5 9-11 The sentence needs rewriting.

We have re-phrased this

P 5

“Obesity is defined as a body mass index (BMI) of ≥ 27.5 kg/m² (or ≥ 30 kg/m² if of White ethnic groups). No country has managed to achieve a sustained decrease in the prevalence of obesity, despite evidence-based clinical and public health guidelines and policies aimed at tackling obesity 2,3.”

4. 5 7-20 You might write a little bit more on the importance of the problem of obesity to the health care system, like what kind of major diseases are related to obesity, what are the consequences for the health care system,...

We have added additional detail on the importance of the obesity problem in terms of associated healthcare conditions and resultant economic cost.

P5

“ Obesity increases the risk of developing several conditions including type 2 diabetes, cardiovascular disease, osteoarthritis, and some cancers 4. The cost attributable to overweight and obesity are substantial. For example, in the UK’s National Health Service (NHS) the cost is projected to reach £9.7 billion per annum, with wider costs to society projected to reach £49.9 billion by 2050 per year^{5,6}. The detrimental effects of excess weight are not restricted to those who meet the BMI threshold of obesity as the increased morbidity is seen in people with any degree of excess adiposity⁷. Accordingly, strategies to prevent obesity or excess weight or adiposity are needed, defined here as obesity and weight-related research.”

Also, make clearer what might be the reasons, why existing evidence did not lead to a reduction in obesity so far – is this because the existing research is not helpful for decision-makers to derive political action from it? Did the research not address the most relevant questions so far (because certain people like patients are caregivers where not involved in setting the research agenda)? Or are there political/economic reasons, why existing evidence could not have been transferred to practice yet?

We have added in a section to address this point:

P5-6

“Presently, the research agenda is mainly driven by the interests and concerns of researchers, or research commissioners. A more transparent, systematic, and collaborative approach involving multiple stakeholders to identify research priorities could accelerate progress..... Policies to prevent obesity typically affect the whole of society, for example fiscal policies or policies restricting the promotion or selling of some goods. Likewise, providing treatment for obesity as part of publicly funded healthcare is contested, and thus questions about research in this area seem to call for a much wider group of stakeholders than patients, carers, and clinicians.”

5. 5 32-41 Make clearer, why also the viewpoints of politicians need to be considered in setting the research agenda for obesity (or overall public health topics) or respectively, why they do not need to be considered in classical JLA approaches.

Also, I do not fully understand the section “there is no obvious patient constituency as we are all at risk of developing overweight” – This argument might also be the case for several other diseases not exclusively caused by genetic reasons (e.g., we are also all at risk to develop a depression or having a heart attack). I do think that for obesity or overweight related health problems there is an obvious patient constituency like for most other diseases as well or I do not fully understand what you are trying to say here.

We have now removed reference to there being no obvious patient group for obesity. We have described why responses were sought from a wider group of stakeholders.

P6

“Policies to prevent obesity typically affect the whole of society, for example fiscal policies or policies restricting the promotion or selling of some goods. Likewise, providing treatment for obesity as part of publicly funded healthcare is contested, and thus questions about research in this area seem to call for a much wider group of stakeholders than patients, carers, and clinicians. As in a previous tobacco control priority setting partnership (PSP) 11, we adapted the JLA approach to incorporate the perspectives of this wider range of stakeholders including people without experience of obesity, policy-makers, charities, and, as for JLA, patients and members of the public with a lived experience of obesity (or related disease), and clinicians 11.”

6. 5 39-41 State, what this modification to the JLA process includes.

We have included a description of the main modification in our approach:

P6

“As in a previous tobacco control priority setting partnership (PSP) 11, we adapted the JLA approach to incorporate the perspectives of this wider range of stakeholders including people without experience of obesity, policy-makers, charities, and, as for JLA, patients and members of the public with a lived experience of obesity (or related disease), and clinicians 11. The objective of this work, as the first prioritisation project in obesity and weight-related research, was to identify unanswered questions across the whole of the field, from basic science through to health policy.”

7. 5,6 52-57, 3-6 As you refer to the JLA process several times, it would be helpful for the readers to provide a short description of the general/classic JLA process and state, what you have adapted/modified. When you write “but including members of...”, it sounds like in the general JLA process these stakeholder groups are normally not involved but this is not true for some of these stakeholders.

We outline the JLA process and reference this. We have stated that we widen the stakeholder group.

P5

“The James Lind Alliance (JLA) priority setting process brings patients, carers, and clinicians together on an equal basis to define uncertainties, consider their importance, and thereby set research priorities 8,9.”

P6

“...we adapted the JLA approach to incorporate the perspectives of this wider range of stakeholders including people without experience of obesity, policy-makers, charities, and, as for JLA, patients and members of the public with a lived experience of obesity (or related disease), and clinicians 11.”

8. 7 6-8 How have you assessed the questions according to the category “answerable by empirical research”. Also, was this the first step and afterwards you categorised the questions further into “already answered” and “unanswered”? If so, writing these three categories in one sentence might be misleading as it can also be understood that you have divided the questions according to these three categories.

We have re-written this section to give more detail about our process.

9. 7 10-13 Did you also have a time limit, like no review older than 3 years (like the JLA suggests it). To which guidelines for assessing the evidence do you refer here? Please also state how you have searched for evidence (e.g., which data bases?)

We have now included details of the limits we used to search the literature for evidence.

P 9

“We then searched the literature using keywords and MeSH terms informed by the questions, to determine if these were areas that were already adequately addressed in the scientific literature.”

P 9

“...systematic reviews published within the last 10 years, with little to no uncertainty;..”

10. 7 12-17 Please describe the grouping and formulation process of the re-search questions more detailed. Did you had a coding system for grouping the research questions? Have you used some guide-lines/frameworks to formulate the research questions?

We had a multi-level grouping system to group questions about similar topics into one category e.g. treatment > behavioural > which diets work?

11. 7 30-34 Describe, how you batched the questions and how many ques-tions were left.

See below and more detail under your point 19.

P 10

“Survey 2 asked respondents to prioritise the unanswered questions gleaned from survey 1, which were sent in batches of about 50 questions to lower the response burden. The questions in each batch covered the whole range of submitted research questions.”

P 16:

“The 149 questions to be taken forward from survey 1 were divided into three batches of up to 50 questions, and randomly assigned to respondent’s survey 2.”

Also, you use the abbreviation RQ only once here without explanation. I would recommend to write research ques-tion instead. Please also state, which online tool you have used for the ques-tionnaire.

We have now use research question in full throughout. We state the online survey software used.

P 8 – survey 1

“The survey was administered online using JISC Online Survey and was piloted with our public co-authors and colleagues in theresearch team, prior to being launched.”

P10, survey 2

The second online survey..... was administered via RedCap, ...”

12. 7 40-45 Say, how many people you have invited and which online-tool you used for the virtual workshop.

We have added this information into the text, see below.

P 14

“We invited 64 stakeholders, 39 people confirmed their acceptance and one person dropped out on the day.”

P11

“The workshop was held via a videoconferencing platform (Zoom), and led by external facilitators from Hopkins van Mil, ...”

13. 7 45-49 Why have you decided to include the extra 10 questions here as well? Please state this clearer.

P11

“Prior to the workshop, participants were given the resulting top 30 questions from survey 2, in addition to a list of 10 other questions from survey 1 that had been asked by more than 10 people (Supplementary Table 3 / Table 4). The difference between the mean ranked scores in survey 2 was subtle. Workshop participants were offered the opportunity to advocate to include any of these extra 10 that they felt should be considered in the workshop to be as inclusive as possible.”

14. .8 section “Stage 3” I would recommend to bring more structure into the de-scription of stage three. First state, what was the aim of this process, then who was involved, how many and what the participants needed to prepare. Afterwards, describe the process used for pri-riority setting (dividing participants into smaller groups, discussion in groups and so on).

We have provided more structure and detail to the methods and results sections, see pages 10-12.

15. 8 53-57 How was the top 10 produced from the fourth highest ranked questions? Can you provide more detail on how the top 10 were produced and how all participants agreed on the final list? (discussion or voting like an example)

We have provided more detail on this in the methods sections, p10-12.

16. 9 14 write the number instead of “Seven-hundred and forty-three”

Changed.

17. 9 19-21 You could provide more information here on the final included questions in stage one. At least indicate the mentioned topic are-as. You could also indicate if some topics areas were more repre-sented, i.e. included more questions than others.

We have now included some additional detail on topic area for the submitted questions and the grouped research questions from survey 1. This information is also presented in Figure 2 a & 2 b.
P 16

“Of the 941 submitted questions most questions concerned: prevention and intervention; mental health; illness, disease and health; and food industry, policy and environment (Figure 2 a). Of the 149 grouped research questions taken forward ‘illness, disease & health’ and ‘metabolism, physiology & appetite’, were the most popular categories and fewer questions concerned age of onset and duration of obesity (Figure 2 b). “

18. . 9 27 write numbers instead of “Four-hundred and five”

Changed

19. 9 26-31 I do not fully understand what you want to say here. Did batching mean that not all persons got the same questionnaire, i.e. the same questions for rating? In the method part I understood that you further combined questions (summarizing similar questions to reduce the overall number of questions) and all participants than got the same questions.

In survey 2 we considered that 149 questions would have been too onerous to ask people to rate. Survey respondents were asked to rate 50 questions, not all 149. The questions covered the whole range of topics. We hope that we have made this clearer. See below

P10

“Survey 2 asked respondents to prioritise the unanswered questions gleaned from survey 1, which were sent in batches of about 50 questions to lower the response burden. The questions in each batch covered the whole range of submitted research questions.”

P 16:

“The 149 questions to be taken forward from survey 1 were divided into three batches of up to 50 questions, and randomly assigned to respondent’s survey 2.”

20. 9 24-32 Provide more information on the rating of the questions (which questions were high ranked and included in stage 3? Are the high ranked topic areas somewhat similar?, Were the participants unanimous in their rating? (provide consensus values),...)

Thank you for your comment. We have provided more detail on the process. See p 10-12.

21. 9 36 Write the number instead of “Sixty-four”

Changed.

22. 9 50 I get a little bit confused here – you write top 10 questions here but wasn’t the aim to rank the top 14 for each group? Why did the group not rank the questions? Due to lack of time?

We have re-written this section. See methods (p 10-12). The aim of the second stage of the workshop was to rank the top 14 questions. In the final session participants were asked to come up with a list of 10 questions in order of priority.

23. 10 15-22 I found this paragraph difficult to understand. You write here for the first time, that you also ranked the questions for survey 1 according to the number of people asking the question. This might confuse the reader as you write in the method part, that you used stage one to gather uncertainties first and used the collected questions for ranking in stage 2 and 3 (no rank-ing in stage 1). Per-haps you can already mention this earlier in the method part that you will com-pare the top ranked questions between each stage.

We noted how many original questions fed into each research question.

This has been added into the methods section

P 10 “We noted how many questions fed into each research question. “

We simplify the section you reference, see below:

P 17.

“Five of the final top 10 questions were among the 10 most frequently submitted questions in survey 1. Seven of the final questions were in the top 10 from survey 2, ranked by mean score.”

24. 11 13-17 I would say these sentences are more the reporting of results than the dis-cussion of them. Provide more information on the last sentence, which topic areas were covered by the top 10 ques-tions. Which public health problems/policy problems are ad-dressed by the ques-tions?

Thank you for your feedback, this section has been removed from the discussion.

At the end of the results section we now reference Table 2 which lists the 10 priorities.

25. 11 32-36 It sounds here like you used the three methods independently to identify and prioritise the top research questions. But it was a process where you reduced the questions in each stage to the final top 10. I would say that it is therefore not really possible to say that the rank-ing of the

questions was not affected by the methods in each stage as the stages were not independent from each other.

Thank you for this comment- we have now removed this.

26. 11 38-45 Please make clearer in the method part, which guidelines you have used and how you have searched for evidence. When you had objective guidelines and a transparent way to exclude research questions that were already answered I would not say that this process is too subjective so you might formulate this a little bit more positive here.

We have now included further detail on how the literature was searched.
In the discussion we also talk through our process.

See response to reviewer 1 comment 6

27. 11 discussion section I am missing here a little bit more discussion related to the research that has been done so far on the issue of obesity compared to the research questions you have identified. Can you for example say, that your research questions represent other interests than previous research?

As we excluded the questions that had already been addressed by research we are taking forward the questions not yet adequately addressed. We hope that this comes through in this paper.

28. 12 45-54 I would mention these topic areas already in the result section.

We now reference Table 2 with the full list of questions at the start of the discussion.

VERSION 2 – REVIEW

REVIEWER	Dean, Caitlin Academisch Medisch Centrum, Obstetrics and Gynecology
REVIEW RETURNED	10-May-2022
GENERAL COMMENTS	Thank you for addressing my comments so thoroughly. My only final comment is that your figures 1 and 2 could do with being produced in a higher resolution as they are difficult to decipher.